# Quantifying protein dynamics and stability in a living organism

Ruopei Feng [1], Martin Gruebele[1,2,3] & Caitlin M. Davis [1,2]

As an integral part of modern cell biology, fluorescence microscopy enables quantification of the stability and dynamics of fluorescence-labeled biomolecules inside cultured cells. However, obtaining time-resolved data from individual cells within a live vertebrate organism remains challenging. Here we demonstrate a customized pipeline that integrates meganuclease-mediated mosaic transformation with fluorescence-detected temperature-jump microscopy to probe dynamics and stability of endogenously expressed proteins in different tissues of living multicellular organisms.

[1] Department of Chemistry, University of Illinois at Urbana-Champaign, Urbana, IL 61801, USA. [2] Department of Physics, University of Illinois at Urbana-Champaign, Urbana, IL 61801, USA. [3] Center for Biophysics and Quantitative Biology, University of Illinois at Urbana-Champaign, Urbana, IL 61801, USA. Correspondence and requests for materials should be addressed to M.G. (email: mgruebel@illinois.edu) or to C.M.D. (email: cmmdavis@illinois.edu)

Tissues are comprised of differentiated cells that can be classified into four broad phenotypes: epithelial, connective, muscle, and nervous. Each phenotype is characterized by a specific morphology, metabolic activity, responsiveness to signals, and overall function. These differences are largely due to modifications in gene expression and the resultant phenotypic specialization of a cell's proteins[1]. In-cell studies have measured heterogeneous properties of proteins at the cellular level, from localization to phase in the cell cycle, but most in-cell studies are conducted in cancer cell lines derived from epithelial tissues[2]. A method for direct quantification of biomolecular stability, interactions, and kinetics in individual cells of differentiated tissues is necessary to reveal the full functionality of biomolecules in their in vivo environment.

Here, we present a customized pipeline (Fig. 1) that combines meganuclease-mediated transformation with fluorescence-detected temperature-jump microscopy to image fast dynamics of biomolecules in living multicellular organisms with single-cell resolution. We demonstrate the method by imaging the folding kinetics and stability of the fluorescence resonance energy transfer (FRET)-labeled glycolytic enzyme phosphoglycerate kinase (PGK) in individual cells of four zebrafish tissues: myocytes, keratinocytes, eye lens, and the notochord. Comparison between in vivo tissues and in vitro experiments shows that all tissue types stabilize proteins over in vitro. The highly crowded lens tissue in particular enhances protein stability and slows folding over all other tissues.

## Results

**Meganuclease-mediated transformation.** Cell-to-cell variation, both spatial and temporal, is always present in populations of cells, but masked by bulk tissue response. Rather than generate uniformly labeled tissues[3] we generate mosaic tissues containing a few labeled cells, thus enabling single-cell studies within the organism. To introduce FRET-labeled protein into single cells of zebrafish, we exploited the large and highly specific recognition sequence of the *I-SceI* meganuclease, which has not been found in any vertebrate genome to date. Our expression cassette (Fig. 1a) contained a promoter, FRET-labeled protein, and polyadenylation signal flanked at both ends by *I-SceI* recognition sites. Our approach relies on nonspecific binding to the host DNA[4,5] and late integration of the transgene to obtain mosaic expression (Fig. 1b). The advantage of this approach is that we can measure and compare individual cells (Fig. 1c).

**Protein folding is probed by temperature perturbation.** The body temperature of poikilothermic organisms is dependent on the surrounding environment. Hence, the temperature inside individual cells of living zebrafish is regulated by the environmental temperature. In vivo thermal stability and kinetics of endogenously expressed FRET-labeled protein is monitored by time-resolved (100 ms) fluorescence microscopy following a small mid-infrared laser-induced temperature jump or resistive heating (Fig. 1d, e). This approach has been established for interrogating protein stability, association, and kinetics in cultured cells[6].

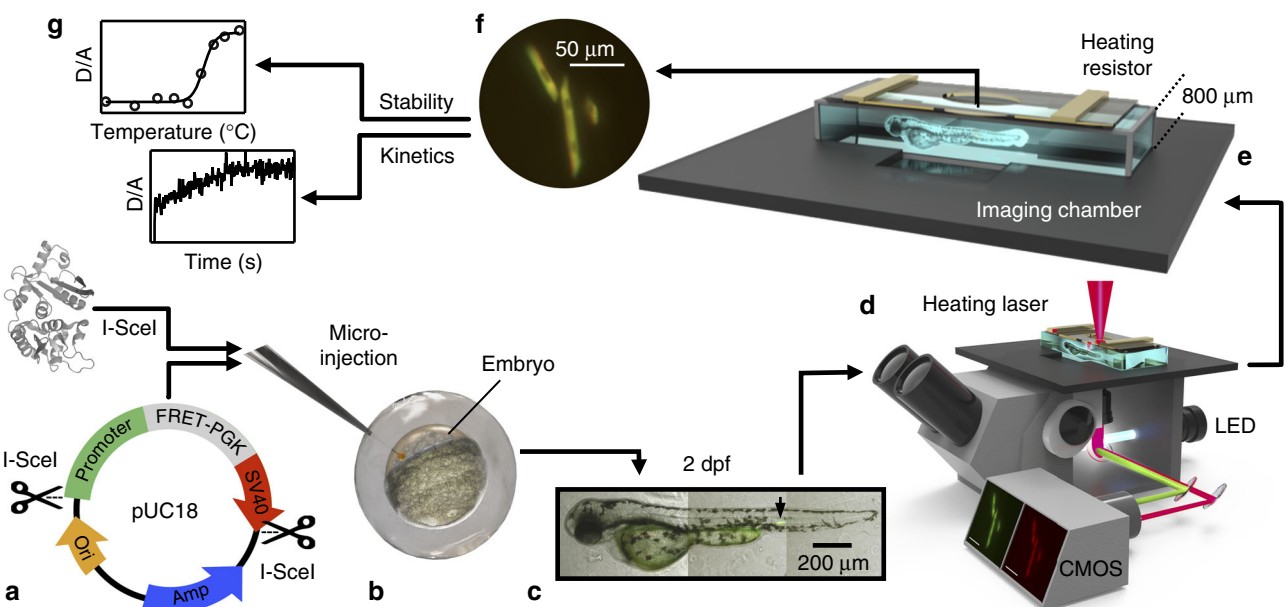

**Fig. 1** A customized pipeline to probe the dynamics and stability of endogenously expressed proteins in different tissues of living zebrafish. **a** The pUC18 transgene cassette is comprised of a tissue-specific zebrafish promoter, FRET-labeled protein, and SV40 polyadenylation signal flanked at both ends by *I-SceI* recognition sites. **b** *I-SceI* meganuclease (PDB ID: 1R7M, rendered using VMD[38]) and the pUC18 transgene cassette are microinjected into single-cell stage zebrafish embryos. **c** Mosaic expression of the FRET-labeled protein is observed in zebrafish larvae 2 days postfertilization (2 dpf). The black arrow points to a single myocyte expressing the FRET-labeled protein. The zebrafish image is a composite of brightfield and fluorescence microscopy images collected at 3 positions under 10× magnification. **d** Schematic of the temperature-jump fluorescence imaging microscope. Individual cells in the living zebrafish are illuminated by a white LED with an appropriate bandpass filter and dichroic for FRET excitation. A heating (infrared) laser initiates a temperature-jump. The two-color fluorescent image is projected onto a CMOS camera capable of recording millisecond time resolution movies of kinetics in the cell. **e** The living 2 dpf zebrafish is placed inside an 800 μm imaging chamber for kinetic and steady-state measurements. Steady-state stability measurements are obtained by applying a voltage to heating resistors, which is dissipated into the sample as heat. **f** Fluorescence microscopy images of individual myocyte cells obtained by overlaying the red and green channel under blue excitation collected at 63× magnification. **g** Representative stability and kinetic measurements extracted from fluorescent images collected during resistive heating and temperature-jump fluorescence microscopy, respectively

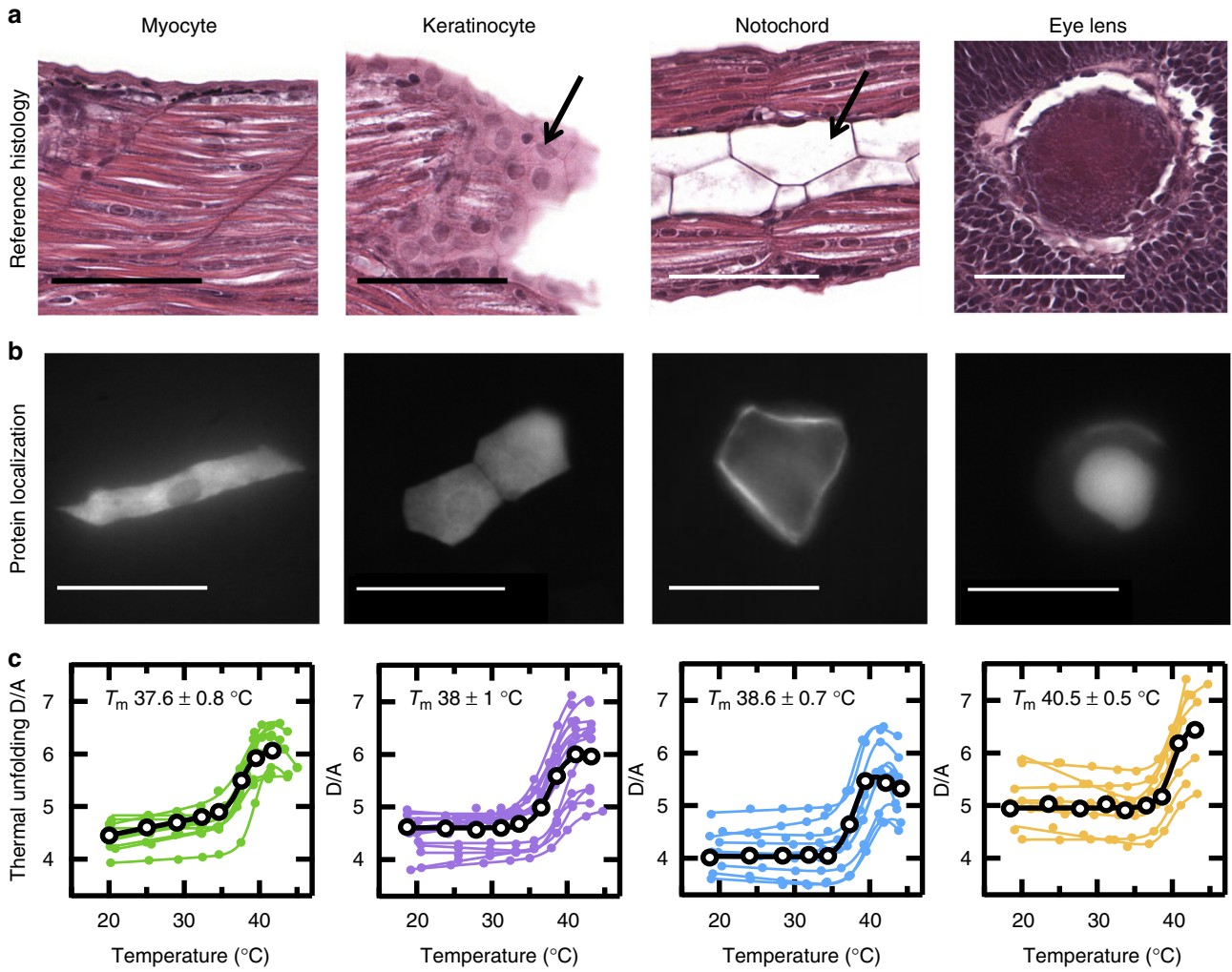

**Fig. 2** FRET-PGK stability and kinetics monitored in zebrafish myocyte, keratinocyte, notochord, and eye lens cells. **a** Hematoxylin and eosin histological virtual slides of 2 day old zebrafish myocyte, keratinocyte, notochord, and eye lens cells. Cell morphology was compared to histology in the zebrafish bio-atlas to confirm nonectopic promoter dependent expression. Myocyte (left) and keratinocyte (left of middle) images from frame http://bio-atlas.psu.edu/zf/view.php?s=528&atlas=49. Notochord (right of middle) image from frame http://bio-atlas.psu.edu/view.php?s=421&atlas=48. Eye lens (right) image from frame http://bio-atlas.psu.edu/zf/view.php?s=530&atlas=49. All scale bars are 50 μm. **b** Promoter dependent tissue expression. EF1α/β-actin promoter localizes PGK expression in myocytes (left). *Krt5* promoter localizes PGK expression in keratinocytes (left of middle). Mouse αA-crystallin promoter localizes PGK expression in the notochord (right of middle). Zebrafish αA-crystallin promoter localizes PGK expression in eye lens cells (right). Fluorescence microscopy images of green channel under blue excitation collected at 63× magnification. All scale bars are 50 μm. Source data are provided as a Source Data file. **c** Thermal stability of PGK in differentiated tissues monitored by fluorescence microscopy. The signal from two-color FRET experiments is reported as donor/acceptor (D/A) ratio. A sigmoidal fit with pre- and post- transition baselines is overlaid on the data. Reported melting temperatures ($T_m$) are averaged over ~8 fish and are presented as mean ± SD. Source data are provided as a Source Data file

The temperature of zebrafish tissues was perturbed in two ways: globally and locally. Globally, resistive heating raised the temperature of the entire imaging chamber, including the zebrafish. This approach was used to monitor protein stability vs. temperature (Fig. 1g). To study protein refolding/unfolding kinetics, a laser temperature jump was used to rapidly perturb the folding equilibrium and monitor protein relaxation. Mid-IR laser light was absorbed uniformly by water inside and outside of an individual cell being imaged, initiating a ~4 °C ~100 ms rise-time temperature-jump (Fig. 1g). The dead time of the kinetics was limited by animal reflex motion at <200 ms. The upper temperature in our experiments was dictated by the temperature tolerance of the zebrafish, as evidenced by cell morphology and zebrafish survival.

**Different cellular environments regulate protein folding**. Protein stability is modulated by a tug-of-war between steric and nonsteric interactions inside cells. Steric interactions are generally stabilizing and arise from crowding due to excluded volume in the cell[7]. Nonsteric interactions are frequently destabilizing and result from transient sticking of protein surfaces to other molecules in the cell[8,9]. Inside cells of different tissues, proteins will encounter different local environments. We hypothesized that different interactions would result in different stability phenotypes for the same protein in different tissues.

To test this hypothesis and assess the performance of our pipeline, we investigated four zebrafish tissues: eye lens, keratinocytes, notochords, and myocytes (Fig. 2). Eye lens cells were selected for a highly crowded internal environment; they have concentrations of structural proteins (crystallins) in excess

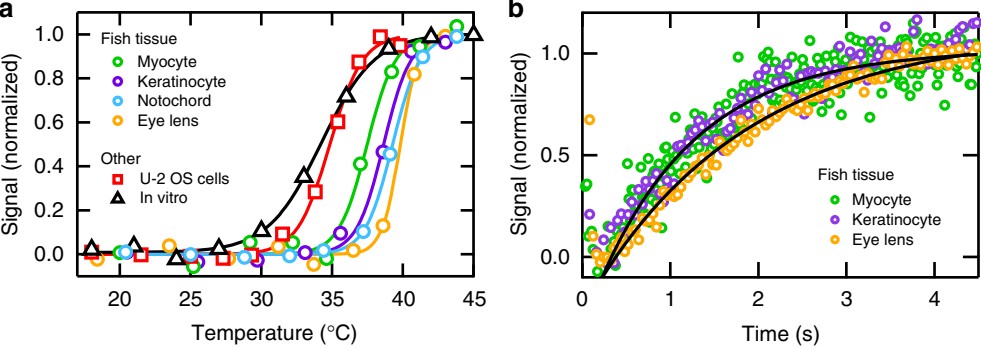

**Fig. 3** Representative thermal denaturation and relaxation kinetics traces of PGK in different tissues. **a** Representative thermal denaturation of PGK FRET monitored by fluorescence microscopy. The signal from two-color FRET experiments is reported as normalized donor/acceptor ratio. The complete denaturation profile between 18 and 45 °C monitored in zebrafish tissue (circles): myocyte (green), keratinocytes (purple), notochord (blue), or eye lens cells (orange); U-2 OS mammalian cells (red squares); or in vitro (black triangles). A sigmoidal fit is overlaid on the data. Source data are provided as a Source Data file. **b** Representative relaxation kinetics of PGK following a temperature jump to $T_m$ observed by fluorescence microscopy. The signal from two-color FRET experiments is reported as normalized donor/acceptor ratio: myocyte (green), keratinocytes (purple), or eye lens cells (orange). A single-exponential fit is overlaid on the data between 200 ms and 4.4 s. Data below 200 ms was not used due to reflexive motion of the zebrafish. Source data are provided as a Source Data file

of 500 mg ml$^{-1}$[10], whereas most cells have concentrations between 50 and 400 mg ml$^{-1}$[11,12]. Keratinocytes were selected because they lie near the animal's surface and thus must achieve allostasis of their internal environment under external stress[13]. The notochord is a centrally located stiff tissue that secrete factors that signal to surrounding cells, providing position and fate information; it was selected because of the unique inflated vacuole that expands to occupy the majority of the cell volume[14]. Finally, myocytes, skeletal muscle cells located in the interior of the zebrafish, were chosen because their internal environment is less crowded and subject to less external stress.

The ATP-synthetase PGK was selected as a model protein because it is ubiquitously expressed in many tissues and it is not an obligatory heat shock protein (Hsp) client: in vitro PGK folds independent of Hsp[15] and in excess Hsp only a small fraction of PGK binds[16]. We designed a new functional mutant of PGK with a low-melting temperature (Fig. 3a, Supplementary Figure 1), allowing us to study stability and folding kinetics at temperatures that are not harmful to zebrafish. PGK is expressed in the cytoplasm and requires a localization tag to direct it to the nucleus or cytoplasmic organelle[17,18]. The even distribution of fluorescence across the cytoplasm in Fig. 2b confirms that PGK uniformly populates the cytoplasm. The fusion protein construct consisted of the mutant PGK labeled on its N and C termini by a green fluorescent protein (AcGFP1) donor (D) and mCherry acceptor (A), respectively. In vitro, the fluorescent proteins unfold at 70 °C, so the labels are stable and changes in donor and acceptor fluorescence are a result of changes in end-to-end distance associated with PGK folding.

**PGK is stabilized in zebrafish tissues**. To measure protein stability and folding in the cell, we measured the donor–acceptor ratio (D/A) as a function of temperature in 9–14 individual cells within each of the three zebrafish tissues (Fig. 2c). The temperature unfolding/refolding of PGK was reversible (Supplementary Figure 2). The thermal unfolding proceeded similarly in all tissues, with greatest stability in eye lens cells: $T_m$ of the protein there was 40.5 ± 0.5 °C (1 SD) vs. 38 ± 1 °C in myocytes, keratinocytes, or the notochord. The increased $T_m$ observed in eye lens cells suggests that their highly crowded crystallin matrix is largely responsible for stabilization of PGK. Indeed, comparison of the tissue-dependent protein stability with in vitro and cultured U-2 OS cells (Fig. 3a) showed the lowest PGK stability,

34.2 ± 0.2 °C, in the dilute aqueous in vitro environment. All fish tissues were at least 3 °C more stable than mammalian U-2 OS cells. Thermodynamic parameters derived from two-state fits of the in-cell thermal melting curves are reported in Supplementary Tables 1–7.

**PGK folding is slowed down in highly crowded eye lens cells**. Although past studies have shown that PGK folding speed is insensitive to environment[15], we hypothesized that the highly crowded eye lens environment may slow PGK folding. Fluorescence microscopy was used to measure folding dynamics following a 100 ms IR laser temperature jump to the melting temperature (Fig. 3b, Supplementary Tables 8–11). Analysis of ~10 fish yielded a relaxation time of $\tau = 1.8 \pm 0.2$ s (1 SD) in eye lens cells and a folding time of $\tau = 1.4 \pm 0.3$ s in myocytes and keratinocytes. Notochords were not tested because of the lower overall fluorescence due to the small cytoplasmic volume of the vacuolated cells. We repeated the same temperature-jump experiments in cultured U-2 OS cells, $\tau = 1.4 \pm 0.3$ s, and in vitro, $\tau = 1.2 \pm 0.2$ s (Supplementary Figure 3). Thus only eye lens cells slow down folding of labeled PGK by a detectable amount, potentially due the elevated intracellular viscosity of the crystallin matrix.

**Discussion**
Our pipeline demonstrates that the different cell environments of different tissues contribute to different protein stability and kinetics phenotypes. The pH of the cytoplasm of eukaryotic cells, like those studied here, is strictly regulated between pH 7.0 and 7.4[19]. Recently, it was shown that differing cytoplasmic material in different organisms can transiently stick to the protein surface and give rise to apparent differences in protein properties[20]. If our observations were a result of transient sticking interactions, removing a surface charge would have a different effect in different cellular environments. We compared the stability of PGK and a surface modified mutant (Supplementary Figure 4) in mammalian U-2 OS cells and zebrafish myocyte and found that the difference in stability between the two PGK variants were the same in the two cell types. This suggests that the different protein stability and kinetic phenotypes observed here are not a result of cytoplasmic sticking.

Macromolecular crowding enhances native state stability relative to the unfolded state by restricting the conformational

freedom of the unfolded protein, and reduces the folding rate slightly[21]. In vitro studies of PGK have shown that its stability and kinetics are more sensitive to crowding than to nonsteric interactions[15]. PGK stability and kinetics were independent of ionic strength between 20 and 200 mM monovalent salt. Low concentrations of cell lysate or lysis buffer stabilized PGK but had no effect on PGK folding kinetics. Whereas, both PGK stability and kinetics were sensitive to macromolecular crowding. Therefore, the increased $T_m$ and slow kinetics observed in eye lens cells are likely due to crowding caused by the high concentration of crystallin.

The bulk of the eye lens is composed of fiber cells, which lose their organelles and nuclei during differentiation. α-, β-, and γ-crystallins are expressed at a high concentration and are organized into a short-range, glass-like order in the cytoplasm to achieve the transparency and refractive properties necessary to focus light on the retina[22]. For the first 3 weeks of development, the zebrafish lens is predominantly composed of β-crystallins[23]. The surfaces of β-crystallins are highly charged, divided into acidic and basic classes, which form homo- or hetero-oligomers. Aside from the macromolecular crowding effects discussed above, β-crystallins may interact directly with PGK through their surface charges, which may be beneficial or counterproductive to protein folding. Our previous studies of PGK in a series of cell-like environments[15] demonstrate that its folding and kinetics are relatively insensitive to such electrostatic interactions.

Muscle tissues contain elongated parallel fibers of proteins for mechanical contraction, whereas keratinocytes are highly regulatory, responding to external stress by protection, secretion, absorption, or filtration. Despite these structural and functional differences, $T_m$ and relaxation time of PGK are similar in myocytes and keratinocytes. PGK in the notochord, which is stiff and dominated by an enlarged vacuole, also shares a similar $T_m$. Therefore, the local cytoplasmic folding environment on these tissues is similar, at least as judged by the probe protein PGK.

In conclusion, we report a customized pipeline that allows us to compare the behavior of the same protein construct in single cells of differentiated tissues of living zebrafish. We envision application of this pipeline for monitoring local differences in protein stability, kinetics, and interaction with other proteins or biomolecules within living transparent organisms. We anticipate that this pipeline will be a useful tool for understanding tissue-specific temperature-sensitive processes, such hyperthermia or photodynamic cancer therapy[24], or gene regulation and function[25].

## Methods

**Protein engineering and expression**. The plasmid for the FRET-labeled yeast phosphoglycerate kinase (yPGK) fusion construct was designed with an in vitro melting temperature of ~35 °C, so that the temperature experiments could be conducted in zebrafish without stressing the fish (tolerance between 6 and 42 °C[26]). Briefly, we used plasmid for an enzymatically active destabilized triple mutant (Y122W/W308F/W333F) of yeast PGK with a melting temperature of ~40 °C[6]. To further destabilize the protein, we developed two destabilized constructs: *Construct 1* with enzymatically active point mutations (F333W (mutating back to wild type), P204H[27]) and *Construct 2* with the enzymatically active point mutations from Construct 1 and loop insertion (89_90insSGGGGAG[28]). A comparison of the sensitivity of the two constructs to transient sticking interactions is provided in the Supplementary Methods. Unless otherwise noted Construct 2 was used in the presented work. The protein was labeled at the N-terminus with AcGFP1 and the C-terminus with mCherry, with a two amino acid linker between the protein and the label. A 6His-tag and thrombin cleavage site were added to the N-terminus of the AcGFP1 to assist with in vitro purification. This gene was cloned between BamHI and NotI in the pDream 2.1 expression vector (GenScript Biotech, Piscataway, NJ), which has both a T7 promoter for expression in *E. coli* and a CMV promoter for expression in mammalian cells.

To obtain the fusion protein for in vitro characterization, BL21-CodonPlus (DE3)-RIPL cells (Invitrogen) were transformed with the plasmid and grown overnight on LB plates with 100 μg ml⁻¹ ampicillin at 37 °C. One colony was selected and grown at 37 °C in 20 mL of LB media containing 100 μg ml⁻¹ ampicillin. The small culture was transferred at ~OD 2 to a 1 L culture with 100 μg ml⁻¹ ampicillin. The 1 L culture was grown to $OD_{600} = 0.60$ and induced overnight at 20 °C with 1 mM IPTG (Inalco, Paris, France). The cells were pelleted and resuspended in 20 mL of lysis buffer (20 mM sodium phosphate, 500 mM NaCl, 10 mM imidazole pH 7.4) per 1 L of cells and lysed by ultrasonification.

The lysate was centrifuged at 10,000 rpm and the supernatant was filtered through a 0.44 μm and then a 0.22 μm syringe filter. The filtered lysate was purified by fast protein liquid chromatography on a 5 mL HisTrap purification column (GE Healthcare, Chicago, IL) using an imidazole elution gradient. The fractions with protein were identified by SDS-PAGE, combined and dialyzed into storage buffer (20 mM sodium phosphate pH 7). Concentrations were determined by measuring the absorbance at 475 nm and using the AcGFP1 extinction coefficient (32,500 cm⁻¹ M⁻¹). The molecular weight was confirmed by low-resolution matrix-assisted laser desorption ionization time-of-flight mass spectrometry. Sample concentrations of 1−5 μM in 20 mM sodium phosphate buffer pH 7 were prepared for in vitro measurements, and studied in the same imaging chambers and under the same experimental conditions as used for in cell and in vivo measurements.

**Cell culture and transfection**. Human bone osteosarcoma epithelial cells (U-2 OS ATCC HTB-96, Manassas, VA) were cultured and grown to 70% confluency in DMEM (Corning, Corning, NY) + 1% penicillin–streptomycin (Corning) + 10% fetal bovine serum (FBS, ThermoFisher Scientific) media. Transfection was performed with Lipofectamine (Invitrogen) following the manufacturer's protocol. At the time of transfection, cells were split and grown on coverslips. Media was changed 6 h after transfection. Cells were imaged 17 h after transfection under the same experimental conditions as in vitro and in vivo measurements. Immediately prior to imaging, cells were placed in imaging chambers filled with optimum (ThermoFisher Scientific) supplemented with 10% FBS.

**Ethics statement**. All experimental procedures in this study were approved by the University of Illinois Institutional Animal Care and Uses Committee protocol #16080.

**Animals**. All experiments were performed on wild-type AB genotype zebrafish (*Danio rerio*) embryos and larvae age 2 dpf (days postfertilization). Embryos and larvae were obtained from breeding of adult zebrafish and were raised at 28.5 °C[29,30]. Adult zebrafish were maintained in a Z-hab mini system (Aquatic habitats, Beverly, MA) fish facility at 28.5 °C on a 14 h:10 h light:dark cycle according to standard protocols. Larvae were anesthetized with 1× tricaine containing fish water prior to embedding in 3% (wt vol⁻¹) methyl cellulose and imaging[31].

**Plasmid constructs**. The pUC18-EF1α/β-Actin-yPGK (7.8 kb) was generated by replacing BamHI and SmaI in the pUC18 vector (GenScript Biotech) with the transgene cassette. The FRET-labeled yPGK gene was removed from the pDream 2.1 expression vector between BamHI and NotI. An *I-SceI* recognition sequence and EF1α/β-Actin promoter[32], frog translation elongation factor 1α (EF1α) enhancer fused to rabbit β-globin intron fused to a zebrafish specific β-actin 2 semiubiquitous promoter, flanked by AgeI, AfeI, and BamHI digest products, respectively, was synthesized by Genscript Biotech. A SV40 polyadenylation sequence and reverse *I-SceI* recognition sequence flanked by NotI, NruI, and SmaI digest products, respectively, was synthesized by Genscript Biotech. The complete cassette flanked by unique restriction enzymes AgeI – *I-SceI* forward – AfeI - EF1α/β-Actin – BamHI – yPGK – NotI – SV40pA – NruI – *I-SceI* reverse – SmaI replaced BamHI and SmaI in the pUC18 vector. The promoter was selected to localize protein expression in the tissue of interest. Other constructs were obtained by inserting different promoters between the AfeI and BamHI site: pUC18-αA-crystallin-yPGK (6.7 kb) with a synthesized 1 kb zebrafish αA-crystallin specific promoter[33], pUC18- Mouse αA-crystallin-yPGK (6 kb) with a synthesized 0.25 kb mouse αA-crystallin promoter that expresses in the notochord[33], or pUC18-krt5-yPGK (8.2 kb) with a zebrafish krt5 epidermis-specific promoter[34]. p5E-krt5 was a gift from Kryn Stankunas (Addgene plasmid #82585). p5E-krt5 between XhoI and BamHI was used as the krt5 promoter insert. All transgene cassettes were verified by sequencing.

**Microinjection of plasmid DNA with meganuclease**. Since we are not breeding the organisms, we term this meganuclease-mediated transformation to distinguish it from meganuclease-mediated transgenesis. Integration of the transgene into the host DNA is late, so few primordial germ cells are expected to integrate the DNA and germline transmission is unlikely.

Microinjector needles, borosilicate glass capillaries (1B100F-4, World Precision Instruments), were prepared in advance using a micropipette puller (P-2000, Sutter Instrument Company) and with a final capillary opening of approximately 10 μm. The microinjection solution, 30 ng μl⁻¹ of pUC18 transgene cassette, 7.5 U of *I-SceI* meganuclease and 1× *I-SceI* buffer to a final volume of 30 μl, was prepared 20 min prior to injections[4,5]. While the microinjection solution was incubating, fish were allowed to spawn by removing a divider separating male and female zebrafish in the tanks. Fertilized eggs were collected immediately after spawning, approximately 20 min later, and collected with a tea strainer in room temperature

0.3× Danieau's solution. Single embryos were transferred and aligned against a microscope slide in a Petri dish using a Pasteur pipette. The microinjection needle was backfilled with 3 µl of microinjection solution and injected into the cytoplasm of one-cell stage embryos using a pressure injector (IM-300, Narishige). The total injection volume was approximately 2 nl per embryo. Embryos were allowed to develop for 48 h in 0.3× Danieau's solution prior to screening for transgene expression and selection for experiments. We observed expression of the reporter gene in approximately 80% of the injected embryos that were free of developmental abnormalities.

**Microscopy setup**. All samples were imaged on a modified Carl Zeiss Axio Observer.A1 microscope body (Zeiss, Thornwood, NY). Imaging chambers were formed by putting together a coverslip onto a standard 1 × 3 in. microscope slide trimmed to 1 × 1 in., separated by an 800 µm spacer (Grace Bio Labs, Bend, OR). The imaging chamber was mounted on a custom-designed Delrin microscope stage and held in place by an aluminum cover plate to provide thermal stability and rapid thermal equilibration (Fig. 1e). An imaging window was drilled into the aluminum plate to enable brightfield microscopy, while minimizing exposed surface area on the imaging chamber.

Steady-state resistive heating was achieved by mounting two resistors onto the aluminum plate. The voltage applied to the resistor is dissipated as heat, which is used to heat the aluminum plate and sample. Steady-state measurements were collected following a 5 min temperature equilibration. Longer equilibrations times were tested to confirm that the equilibration time was sufficient.

The sample was excited by passing light from a white UHP-T2 LED head (Prizmatix, Southfield, MI) through an appropriate bandpass filter and dichroic. An ET470/40× bandpass filter (Chroma, Bellows Falls, VT) and T495lpxt dichroic (Chroma) were used for AcGFP1 and FRET excitation. An ET580/25× bandpass filter (Chroma) and T600lpxr dichroic (Chroma) were used for mCherry excitation. The excitation beam was focused onto the sample by a 63×/0.75 NA LD Plan-Neofluar objective (Zeiss). The emission was collected in epifluorescent mode, passed through an ET500lp filter (Chroma) and split into two channels by a T600lpxr dichroic (Chroma). Images were collected with a LT225 NIR/SCI CMOS detector (Lumenera, Ottawa, Canada) at 16–60 ms integration times. The detector was controlled using LabView (National Instruments, Austin, TX).

**Temperature calibration**. Two methods were used to calibrate the temperature. First, the temperature was calibrated from above the slide using a thermocouple. A voltage is applied to the resistor, which is dissipated as heat, and used to heat the sample. The voltages necessary to achieve ~3 °C temperature steps were used for experiments. Second, as a check of the external thermometer, the temperature-dependent quantum yield of mCherry was used to monitor temperature directly in the sample (in vitro, in cell, or in vivo)[17].

**Temperature jumps**. A tailored continuous wave laser was used to rapidly perturb the folding equilibrium on a timescale faster than the molecular dynamics of interest. Fluorescence microscopy was then used to probe the reaction as it relaxes to the new folding equilibrium at the higher temperature. The observed relaxation rate is the sum of the forward and reverse reactions (unfolding and folding): $k_{obs} = k_{folding} + k_{unfolding}$. At final jump temperatures below $T_m$ the reaction is dominated by the folding rate ($k_{folding}$) and at final jump temperatures above $T_m$ the reaction is dominated by the unfolding rate ($k_{unfolding}$).

A computer controlled continuous wave 2.2 µm mid-infrared diode laser (m2k-laser GmbH, Breisgau, Germany) aligned to 90° to the normal of the sample was focused on a small 400 µm full width half max spot around the cell of interest to generate the temperature-jump (T-jump) (Fig. 1d). Images were collected at 60 or 25 Hz frame rates with 16 and 40 ms integration times, respectively. The relaxation of the protein to its new equilibrium was measured for 6 s prior to allowing the sample to relax back to the original temperature. The magnitude of the T-jump was calculated using the temperature-dependent quantum yield of mCherry[17]. Considerations for applying this method to other organisms are provided in the Supplementary Methods.

**Image processing**. Fluorescent images of fish cells were analyzed in MATLAB (Mathworks, Inc.). Owing to the low background in the images, we were able to separate the cells from the background using a simple image binarization with a threshold proportional to the brightness of the background. Each cell outline was calculated based on the green channel image and applied to both green and red channel images. The donor (D) and acceptor (A) intensities were calculated by averaging the fluorescence intensities of the cell in the two channels. We have

$$D = \frac{1}{N}\sum_P i_g^p,$$ (1)

$$A = \frac{1}{N}\sum_P i_r^p,$$ (2)

where $P$ is a pixel within the cell outline, $N$ is the number of pixels within the cell outline and $i_g$ and $i_r$ are the intensity of a pixel in green channel and red channel,

respectively. For the cases when multiple cells were present in one image, we manually created a polygon mask for each cell and used it to segment and analyze individual cells.

**Analysis of thermodynamic data**. Equilibrium measurements were collected between room temperature and 42 °C in ~4 °C steps. Thermodynamic data is plotted as the ratio of the donor (D) and acceptor (A) intensities (D/A) vs. temperature. Using the van't Hoff relationship, the differential melting curve can be robustly fit to three parameters with no sensitivity to the pre- and post-transition baseline[35,36]. This method also accurately predicts melting temperature ($T_m$) for truncated data sets. Hence, we fit the first derivative of the thermal melts (Supplementary Figure 1) to an apparent two-state equilibrium model[35]:

$$\frac{d(signal)}{dT} = Af(1-f)(T^2),$$ (3)

where $d(signal)/dT$ is the algebraic derivative of the D/A signal, $A$ is a scaling factor and $f$ is the fraction of denatured protein. $f$ is related to the melting temperature ($T_m$) and van't Hoff enthalpy ($\Delta H_{vH}$) by the equilibrium constant ($K$) for the unimolecular reaction between native ($N$) and denatured ($D$) states:

$$K = \frac{D}{N} = \frac{f}{1-f},$$ (4)

$$K = \exp\left[\frac{\Delta H_{vH}}{R}\left(\frac{1}{T_m} - \frac{1}{T}\right)\right].$$ (5)

Fits to Eq. (5) were accepted if the error in each of the parameters was an order of magnitude lower than the fit. The data analysis was performed in IGOR PRO (WaveMetrics, Lake Oswego, OR).

Iglewicz and Hoaglin's robust test (two-sided) for multiple outliers was used to identify potential outliers in the thermal melts[37]. Using a modified $Z$ score ≥ 3.5, no cells were identified as potential outliers in the myocyte data and two cells were identified as potential outliers in each of the keratinocyte and eye lens cells. Further inspection revealed abnormalities in these fish, high background in the keratinocyte data and abnormal expression in the eye lens cells. These cells are eliminated from the reported results but included in Supplementary Tables 1–7. Additional keratinocyte and eye lens measurements were collected so that results come from eight different fish. Reported results are the average of the mean of 14 keratinocyte, 10 myocyte, 9 eye lens, and 19 U-2 OS cells.

**Analysis of kinetic data**. To generate the kinetic data, the D/A is plotted against time. The kinetic data was fit to a single exponential[17]:

$$S(t) = S_0 + A_1 e^{(t/\tau_{obs})^\beta},$$ (6)

where the exponential accounts for the sample kinetics, $S_0$ is an offset, $A$ is the preexponential factors, $\tau$ is the relaxation lifetime, and $\beta$ is the stretched exponential factor. $\beta = 1$ for 2-state folders, but may differ from one if there is multistate folding or in an inhomogeneous environment. $\beta = 1$ for the PGK data presented here. The data are fit over the interval from 200 ms to 4 s. The data analysis was performed in IGOR PRO. Reported results are the average of the mean of 10 keratinocyte, 12 myocyte, 10 eye lens, and 13 U-2 OS cells.

## Data availability
Complete tables of the equilibrium thermodynamic and kinetic parameters of myocyte, keratinocyte, eye lens, notochord, and U-2 OS cells are available in Supplementary Tables 1–11. The source data underlying Fig. 2b, c, 3, and Supplementary Figures 1–4 are provided as a source data file. Additional data that support the findings of this study are available from the corresponding authors upon reasonable request.

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

## Acknowledgements

This work was supported by the National Science Foundation (NSF) grant NSF MCB 1803786 to M.G. C.M.D. and R.F. were supported in part by the Physics Frontier Center for the Physics of Living Cells funded by NSF PHY 1430124. Figure 1 was designed by the authors and rendered with support from the Beckman Institute Imaging Technology Group (University of Illinois), whose support is gratefully acknowledged. We would also like to acknowledge http://zfatlas.psu.edu/ (NIH grant 5R 24RR01744), Jake Gittlen Cancer Research Foundation, and PA Tobacco Settlement Fund for use of the zebrafish histology slides in Fig. 2.

## Author contributions

C.M.D. engineered the protein and plasmid DNA, and supervised R.F. R.F. bred and maintained the zebrafish. C.M.D. and R.F. designed and implemented the instrumentation and software, performed the experiments, analyzed the data, and wrote the paper. M.G. conceived the experiment, analyzed data, supervised C.M.D. and R.F., and revised the paper.

## Additional information

**Competing interests:** The authors declare no competing interests.

