## [Peer Review File · Nature Communications]

Reviewers' comments:

Reviewer #1, an expert in temperature jump and protein folding (Remarks to the Author):

New temperature jump method focused on single cell measurement and its application to the zebrafish system are demonstrated in this manuscript. Basically it appeared to be somewhat exciting but simultaneously I felt that the application might be quite limited. They interpreted the increase of T_m and time constant in eye is due to the crowding effects. But we do not know the intra-cellular location of the expressed PGK, so we do not know how proteins are crowded around PGK. Some protein may interact with PGK directly and stabilize or destabilize it. Local pH might be different, and intracellular environment is quite heterogeneous between the intracellular locations. Since only a single data which shows the stabilization of PGK is presented, it is hard to evaluate the reliability of data and also to discuss the nature of interprotein interaction in the cell. In methodological view point, there is not much essential breakthrough in temperature jump technique, which must be described more thoroughly by Dr. Gruebelle including its innovative points as well as its limitations. Therefore I recommend to accumulate the data further to address the above basic questions and resubmit the manuscript again to Nature Communication.

Reviewer #2, an expert in FRET and protein biophysics (Remarks to the Author):

The manuscript by Feng et al. describes what appears to be the first quantitative analysis of protein stability and folding kinetics in distinct cell types in a living organism. Their protein model system, which contains fluorescent proteins that exhibit FRET in the folded state, is carefully and appropriately engineered for their studies as is their method of transfection and integration into the genomes of a variety of different cell types. More precision experiments enabling cell-type specific expression of their protein folding/unfolding sensor could have been achieved through genetic engineering of the animal under study, but this would be both laborious and it would restrict their studies to just one type of organism or cell type. The elegance of their system is that it can be employed ubiquitously.

Although there are clear differences in unfolding and folding kinetics of their protein model system in lens tissues compared to other tissues, which is important to demonstrate proof-of-principle of the approach, it is not immediately obvious that the general reader will be blown away by these findings. That being said, the authors do appear to be the first to show such studies are feasible and the measurements are robust and rigorously assessed. The manuscript is also well written and appropriately referenced (in my view). Thus, the work certainly warrants publication. Given the likelihood that this research sets the precedent for many downstream investigations to come that will build upon and leverage these proof-of-principle studies, I would also think this work suitable for publication in Nature Communications.

Reviewer #3, an expert in in vivo protein folding and thermodynamics (Remarks to the Author):

In this manuscript the authors demonstrate a new, quantitative method to address the question on protein thermodynamic ability as well as folding kinetics in live cells in multicellular organisms. This is indeed an important contribution to the field that strives to achieve a molecular understanding for protein function and dysfunction in the native habitat of the protein. The method is based on state of the art instrumentation and sample handling. The authors have studied folding kinetics and thermodynamic stability of the well studied FRET-labelled PGK in buffer, cultured cells and three different cell types in live zebrafish. The PGK variant used here is tuned to a melting point at a temperature that are tolerable for the organism, enabling thermal unfolding without harming the surrounding tissue. This is a very strong and well researched study and I recommend publication in Nature Communications after addressing a few minor points.

(i) The authors discuss the competing stability modulating factors in the cell; steric, volume exclusion interactions that stabilise the protein and transient interactions that destabilise the protein. In all cell types, both these types of interactions should be present and the most dominating will determine the outcome. Here the authors find minute differences between the cell-types. Actually, the differences is surprisingly small, given the expected differences in overall cytosolic macromolecular concentration. This could be by chance: the transient and the steric interactions may be balanced in the very high concentrated, but homogenous eye lens cells. The bench marking of the method would strengthen even further if a surface modulated variant of the FRET-PGK was tested, where the surface modulation would mainly affect the transient interactions while leave the steric interactions less affected.

(ii) The eye lens cells are densely packed with crystallin, which makes them attractive as test environment, but the cells also stands out as an outlier as the protein composition relatively homogenous, with the absolute vast majority of proteins being a couple of sub classes of crystallins. This may be both beneficial and counter productive, and the authors should discuss the complication of using this cell type.

(iii) The differences in both stability and kinetics are, as mentioned above, very minute. Could the effects come from variations in other parameters than protein concentration, such as pH, ionic strength or ionic composition (small soluble ions or large charge complexes)? Is the PGK model system independent on these matters? I think the paper would benefit from a discussion on such matters as well.

(iv) For the unfolding kinetics in figure 2d the amplitudes and the burst phase of the signal differs between the eye lens data and the other cell types. Is this difference reproducible, and if so, is there an explanation for that? Is it just differences in dead time events? And if so, can a such small difference in relaxation time explain such large difference?

Reviewed by Jens Danielsson

Response to Reviewers:

Reviewer #1:

(1) They interpreted the increase of T_m and time constant in eye is due to the crowding effects. But we do not know the intra-cellular location of the expressed PGK, so we do not know how proteins are crowded around PGK. Some protein may interact with PGK directly and stabilize or destabilize it. Local pH might be different, and intracellular environment is quite heterogeneous between the intracellular locations.

Author reply: Excellent point. Please see response to the similar point made by Reviewer #3 (iii).

(2) Since only a single data which shows the stabilization of PGK is presented, it is hard to evaluate the reliability of data and also to discuss the nature of interprotein interaction in the cell.

Author reply: As stated in the caption to Figure 2, equilibrium measurements were averaged over ~ 8 fish and kinetic measurements were averaged over ~ 10 fish. Additional details on the absolute numbers and fits of each cell type are included in the SI on page S-5 and Tables S1-S3 and Tables S5-S7.

To clarify the number of measurements in the main text and show the range of results obtained in different cells of the same type, we have updated Figure 2c to include all of the equilibrium melts:

2 (c) Thermal stability of PGK in differentiated tissues monitored by fluorescence microscopy. The signal from two-color FRET experiments is reported as donor/acceptor (D/A) ratio. A sigmoidal fit with pre- and post-transition baselines is overlaid on the data. A representative equilibrium melt is highlighted in black. Reported melting temperatures (T_m s) are averaged over ≈ 8 fish.

(3) In methodological view point, there is not much essential breakthrough in temperature jump technique, which must be described more thoroughly ... including its innovative points as well as its limitations.

Author reply: We agree, the technical breakthrough here is not the temperature-jump technique, but the ability to collect protein stability and kinetics measurements in individual cells of a living organism. We have expanded the discussion of practical considerations for temperature jump measurements in living organisms in the Temperature jumps section in the SI page S-4:

“In principle this approach can be applied to measure protein dynamics in any transparent living organism. Objectives must be selected with a long free working distance for thicker specimen and with an appropriate magnification for the cell type of interest. Our temperature-jump approach requires objectives that are not immersed in fluid, because heat will dissipate more quickly through the immersion fluid (oil, water, glycerin) than the air.

The thickness of the sample can impact (1) the duration of the T-jump and (2) the uniformity of heating through the sample:

- (1) The duration of the T-jump is determined by the by thermal diffusion out of the heated volume. The advantage of the programmable laser is that the temperature profile can be adjusted so that the temperature following the jump is constant within ± 0.25 °C. The power density at the sample was controlled by a TTL voltage input provided by LabView (National Instruments, Austin, TX).
- (2) The sample heating can change in the x, y, or z direction. Aligning the pump laser 90° normal to the sample ensures that there is no x or y drift in the heated volume with changes in z. The sample heating is not uniform with z because of the thickness of the sample.(a) This means that cells found at different heights in the zebrafish would experience a different size temperature jump. To overcome this, we designed three laser output power profiles at 0, 135 or 270 μm above the coverslip to achieve a consistent ~ 4 °C temperature-jump.”

(a) Kubelka, J. Time-resolved methods in biophysics. 9. Laser temperature-jump methods for investigating biomolecular dynamics. *Photochem. Photobiol. Sci.* **8**, 499–512 (2009).

Reviewer #2

Although there are clear differences in unfolding and folding kinetics of their protein model system in lens tissues compared to other tissues, which is important to demonstrate proof-of-principle of the approach, it is not immediately obvious that the general reader will be blown away by these findings. That being said, the authors do appear to be the first to show such studies are feasible and the measurements are robust and rigorously assessed. The manuscript is also well written and appropriately referenced (in my view). Thus, the work certainly warrants publication. Given the likelihood that this research sets the precedent for many downstream investigations to come that will build upon and leverage these proof-of-principle studies, I would also think this work suitable for publication in Nature Communications.

Author reply: We thank reviewer #2 for the positive comments, and explained the origin of folding differences observed in more detail, see replies to reviewers #1 and #3.

Reviewer #3 Jens Danielsson:

(i) The authors discuss the competing stability modulating factors in the cell; steric, volume exclusion interactions that stabilize the protein and transient interactions that destabilize the protein. In all cell types, both these types of interactions should be present and the most dominating will determine the outcome. Here the authors find minute differences between the cell-types. Actually, the difference is surprisingly small, given the expected differences in

overall cytosolic macromolecular concentration. This could be by chance: the transient and the steric interactions may be balanced in the very high concentrated, but homogenous eye lens cells. The bench marking of the method would strengthen even further if a surface modulated variant of the FRET-PGK was tested where the surface modulations would mainly affect the transient interactions while leave the steric interactions less affected.

Author reply: We thank Dr. Danielson for his suggestion. To address this comment, we have designed a version of our FRET-PGK with the loop insertion removed, and added new data. The following discussion, figure, and data were added to the main text and SI.

Added discussion to main text on page 5:

“The pH of the cytoplasm of eukaryotic cells, like those studied here, is strictly regulated between pH 7.0-7.4 (1). Recently, it was shown that differing cytoplasmic material in different organisms can transiently stick to the protein surface and give rise to apparent differences in protein properties(2). If our observations were a result of transient sticking interactions, removing a surface charge would have a different effect in different cellular environments. We compared the stability of PGK and a surface mutant (SI and Figure S4) in mammalian U-2 OS cells and zebrafish myocyte and found that the difference in stability between the two PGK variants were the same in the two cell types. This suggests that the different protein stability and kinetic phenotypes observed here are not a result of cytoplasmic sticking.”

Added to Protein engineering and expression on SI page S-1:

“To further destabilize the protein, we developed two destabilized constructs: Construct 1 with enzymatically active point mutations (F333W (mutating back to wild type), P204H) and Construct 2 with the enzymatically active point mutations from Construct 1 and loop insertion (89_90insSGGGGAG). Unless otherwise noted Construct 2 was used in the presented work.”

Added a new section Sensitivity to transient interactions on SI page S-6:

“To test the sensitivity of PGK to transient sticking interactions we designed a surface modulated variant of FRET-PGK (Construct 1 above) where the loop insertion was mutated back to wildtype. The flexibility of the N-terminal where the loop is located is unaffected by loop insertion (3), so removal of the loop will not impact the sensitivity of PGK to macromolecular crowding. On the other hand, the loop is located at the surface of PGK and contains a polar serine group, so removal of the loop could impact the way PGK interacts transiently with the environment.

We compared the stability of FRET-PGK with and without the loop in two different cell types, mammalian U-2 OS cells and zebrafish myocytes (Table S1, S5-7). If transient interactions arising from the different local environments contribute to in-cell stability, we would expect the difference between the stability of the two proteins in U-2 OS cells to be different from myocytes. Instead, we observe that the onset of unfolding of the two

constructs differs by ≈ 4 °C in both cell types (Table S1, S5-7, Figure S4). This demonstrates that PGK is relatively insensitive to transient interactions.”

Figure S4. Representative thermal denaturation of FRET-PGK without (Construct 1) and with (Construct 2) the loop mutation monitored by fluorescence microscopy. The signal from two-color FRET experiments is reported as donor/acceptor ratio. The denaturation profile between 18–45 °C monitored in U-2 OS cells **(A)** and zebrafish myocyte **(B)**. Each panel is globally fit to a sigmoid with a different T_m , the same ΔH , and the same pre- and post- transition baselines. Note that in (B) only the onset of unfolding is observed for construct 1.

(1) Madshus, I. H. Regulation of intracellular pH in eukaryotic cells. *Biochem. J.* 250, 1–8 (1988).

(2) Mu, X. *et al.* Physicochemical code for quinary protein interactions in *Escherichia coli*. *Proc. Natl. Acad. Sci.* 114, E4556–E4563 (2017).

(3) Collinet, B., Garcia, P., Minard, P. & Desmadril, M. Role of loops in the folding and stability of yeast phosphoglycerate kinase. *Eur. J. Biochem.* 268, 5107–5118 (2001).

(ii) The eye lens cells are densely packed with crystallin, which makes them attractive as test environment, but the cells also stand out as an outlier as the protein composition is relatively homogenous, with the absolute vast majority of proteins being a couple of sub classes of crystallins. This may be both beneficial and counter productive, and the authors should discuss the complication of using this cell type.

Author reply: As suggested, we have added a more in-depth description of the eye lens cells and how this may effect protein folding to pg. 5:

“The bulk of the eye lens is composed of fiber cells, which lose their organelles and nuclei during differentiation. α -, β -, and γ -crystallins are expressed at a high concentration and are organized into a short-range, glass-like order in the cytoplasm to achieve the transparency and refractive properties necessary to focus light on the retina (1). For the first three weeks of development, the zebrafish lens is predominantly composed of β -crystallins (2). The surfaces of β -crystallins are highly charged, divided into acidic and basic classes, which form homo- or hetero-oligomers. Aside the from macromolecular crowding effects discussed above, β -crystallins likely interact directly with PGK through their surface charges, which may be beneficial or counterproductive to protein folding. Our

previous studies of PGK in a series of cell-like environments (3) demonstrate that its folding and kinetics are relatively insensitive to such electrostatic interactions.”

We also conducted additional measurements on another unique cell type, the notochord. Notochord data was added to Figure 2, Figure S1, and Table S4. A short description of the cell type was added to the main text pg 3:

“The notochord is a centrally located stiff tissue that secretes factors that signal to surrounding cells, providing position and fate information; it was selected because of the unique inflated vacuole that expands to occupy the majority of the cell volume (4).”

(1) Delaye, M. & Tardieu, A. Short-range order of crystallin proteins accounts for eye lens transparency. *Nature* 302, 415–417 (1983).

(2) Greiling, T. M. S., Houck, S. A. & Clark, J. I. The zebrafish lens proteome during development and aging. *Mol. Vis.* 15, 2313–2325 (2009).

(3) Davis, C. M. & Gruebele, M. Non-steric interactions predict the trend and steric interactions the offset of protein stability in cells. *ChemPhysChem* 19, 2290–2294 (2018).

(4) Kathryn Ellis, B. D. H. and M. B. The vacuole within How cellular organization dictates notochord function. *Bioarchitecture* 3, 64–68 (2013).

(iii) The differences in both stability and kinetics are, as mentioned above, very minute. Could the effects come from variations in other parameters that protein concentrations, such as pH, ionic strength or ionic composition (small soluble ions or large charged complexes)? Is the PGK model system independent on these matters? I think the paper would benefit from a discussion on such matters as well.

Author reply: We thank both Reviewer 1 and Dr. Danielsson for pointing out these important alternative effects.

The difference between *in vitro* or cultured cell lines and zebrafish is not minute. To better highlight this we have moved Figure S1a to the main text Fig 3a. An additional sentence was added to clarify the stability difference in fish over U-2 OS cells:

“Indeed, comparison of the tissue-dependent protein stability with *in vitro* and cultured U-2 OS cells (Figure 3a) showed the lowest PGK stability, 34.2 ± 0.2 °C, in the dilute aqueous *in vitro* environment. All fish tissues were at least 3 °C more stable than mammalian U-2 OS cells.”

Two sentences discussing cellular localization were added to the main text on page 4:

“PGK is expressed in the cytoplasm and requires a localization tag to direct it to the nucleus or cytoplasmic organelle (1, 2). The even distribution of fluorescence across the cytoplasm in Figure 2B confirms that PGK uniformly populates the cytoplasm.”

A sentence about the pH of the cytoplasm was added to the beginning of a new discussion of cytoplasmic sticking on page 5:

“The pH of the cytoplasm of eukaryotic cells, like those studied here, is strictly regulated between pH 7.0-7.4 (3). Recently, it was shown that differing cytoplasmic material in different organisms can transiently stick to the protein surface and give rise to apparent differences in protein properties (4). If our observations were a result of transient sticking interactions, removing a surface charge would have a different effect in different environments. We compared the stability of PGK and a surface mutant (SI and Figure S4) in mammalian U-2 OS cells and zebrafish myocyte and found that the difference in stability between the two PGK variants were the same in the two cell types. This suggests that the different protein stability and kinetic phenotypes observed here are not a result of cytoplasmic sticking.”

The discussion of the sensitivity of PGK to non-steric interactions: ionic strength and ionic composition has been expanded on page 5.

“*In vitro* studies of PGK have shown that its stability and kinetics are more sensitive to crowding than to non-steric interactions (5). PGK stability and kinetics were independent of ionic strength between 20-200 mM monovalent salt. Low concentrations of cell lysate or lysis buffer stabilized PGK but had no effect on PGK folding kinetics. Whereas, both PGK stability and kinetics were sensitive to macromolecular crowding.”

(1). Dhar, A. *et al.* Protein stability and folding kinetics in the nucleus and endoplasmic reticulum of eucaryotic cells. *Biophys. J.* 101, 421–430 (2011).

(2). Tai, J., Dave, K., Hahn, V., Guzman, I. & Gruebele, M. Subcellular modulation of protein VlsE stability and folding kinetics. *FEBS Lett.* 590, 1409–1416 (2016).

(3) Madshus, I. H. Regulation of intracellular pH in eukaryotic cells. *Biochem. J.* 250, 1–8 (1988).

(4) Mu, X. *et al.* Physicochemical code for quinary protein interactions in *Escherichia coli*. *Proc. Natl. Acad. Sci.* 114, E4556–E4563 (2017).

(5) Davis, C. M. & Gruebele, M. Non-steric interactions predict the trend and steric interactions the offset of protein stability in cells. *ChemPhysChem* 19, 2290–2294 (2018).

(iv) For the unfolding kinetics in figure 2d the amplitudes and the burst phase of the signal differs between the eye lens data and the other cell types. Is this difference reproducible, and if so, is there an explanation for that? Is it just differences in dead time events? And if so, can a such small difference in relaxation time explain such large difference?

Author reply: The spike at fast time was due to fish reflex in response to the T-jump. This limits our dead-time of the kinetics measurement to about 200 ms. ~10 measurements of each cell type were conducted to check reproducibility. We clarified this in the text on pg 4 and we replotted the data more clearly in Figure 3, so the differences in kinetics can be compared directly. Only the eye lens cell shows slightly slower kinetics, possibly due to the increased viscosity of the crystallin matrix:

Pg 4:

“Analysis of ≈ 10 fish yielded a relaxation time of $\tau = 1.8 \pm 0.2$ s in eye lens cells and a folding time of $\tau = 1.4 \pm 0.3$ s in myocytes and keratinocytes. Notochords were not tested because of the lower overall fluorescence due to the small cytoplasmic volume of the vacuolated cells.”

REVIEWERS' COMMENTS:

Reviewer #3 (Remarks to the Author):

The authors have thoroughly and clearly addressed my initial concerns and in my opinion this manuscript could be published in its present form.

Jens Danielsson

REVIEWERS' COMMENTS:

Reviewer #3 (Remarks to the Author):

The authors have thoroughly and clearly addressed my initial concerns and in my opinion this manuscript could be published in its present form.

Jens Danielsson

The referees have not raised any additional concerns.